# CASR: Refining Action Segmentation via marginalizing frame-level causal relationships

## Abstract

Integrating deep learning and causal discovery has increased the necessity for a causal relationship between frames as evidence for explainability in Temporal Action Segmentation (TAS) tasks. However, frame-level causal relationships apparently emerge noise outside the segment, making it infeasible to suggest macro action relationships through frame relationships. To bridge this research gap, we propose a method of marginalizing frame-level noise relationships and introduce a Causal Abstraction Segmentation Refiner (CASR) to enhance the segmentation ability. Specifically, we retain all cross-segment relationships while discarding all inter-segment relationships over the frame-level model, satisfying a consistent mapping of causal abstraction in terms of action semantics from frames to segments. Then identify whether each frame belongs to the corresponding segment by contrastive learning, enhancing the segmentation performance. In addition, we propose a loss function to evaluate the causal interpretability of segmentation results. Extensive experimental results on mainstream datasets indicate that our method significantly surpasses existing various methods in action segmentation performance, and in causal explainability. This generalization performance will make CASR an effective tool for boosting the existing approaches for temporal action segmentation. Our code is available in: `https://anonymous.4open.science/r/CASR`.

## 1 Introduction

Temporal action segmentation (TAS) aims to identify and segment actions, has attracted a lot of attention in the fields of human-computer interaction (Ma et al., 2021; Zhai et al., 2023; Luan et al., 2021), surveillance (Hossain et al., 2019) and security(De Rossi et al., 2021; Liu et al., 2023). Moreover, with the fusion of explainable AI (Xu et al., 2019; Angelov et al., 2021) and deep causal discovery (Deng et al., 2022; Berrevoets et al., 2023), it has become a mainstream choice to infer evidence of model decisions by identifying fine-grained causal relationships between frames, such as from Learning temporal causal relationships in video frames (Zhang et al., 2021), generating causal video summaries (Huang et al., 2022).

Nonetheless, the analysis of precise causal relationships within video content via direct frame-level modeling presents a formidable challenge. As shown in Figure 1(a), as the micro-variables within a system, the relationships between frames are intricate. Some frames within one action segment may exhibit additional correlations with frames from other action segments, making it difficult to cluster macro segments directly through frame-level causal relationships. On the contrary, we found that the adjacency matrix obtained when constructing SEM directly with action segments as units is more regular. As shown in Figure 1(b), the similarity between any two segments is far less than its self-loop similarity, and the segment-level matrix has the ability to reflect the action relationships in the frame-level matrix.

Simultaneously, our investigation has revealed a dearth of explainability in existing studies. Prior related works focus on capturing temporal features of different periods or positioning segmented frames (Zhai et al., 2022) from different supervised learning perspectives and backbones. For instance, MS-TCN++ (Li et al., 2020) and MS-TCN (Farha & Gall, 2019) are grounded in multi-stage TCN

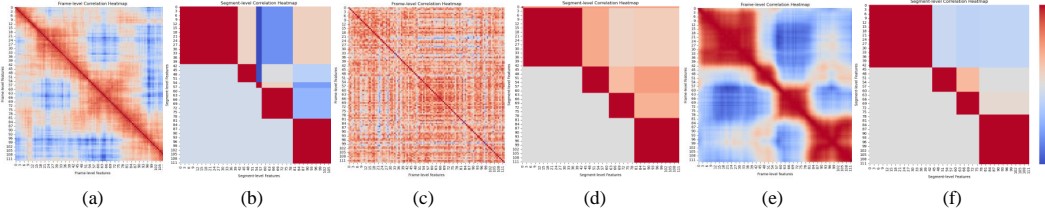

Figure 1: The similarity of the frame and segment feature vectors from 5 action segments in GETA dataset (Fathi et al., 2011). The cooler the color, the lower the similarity; the warmer the color, the higher the similarity. (a) comes from the frame-level of source data. (b) comes from the segment-level of source data. (c), (d), (e), (f) come from the features of frame-level and segment-level after MS-TCN++ and MS-TCN++ with CASR respectively.

architectures designed to capture time-domain features of varying durations, while ASRF (Ishikawa et al., 2021) adds an additional boundary probability branch to glean segmentation point information. However, taking MS-TCN++ as an example, it becomes evident that while the similarity between its video segments after the action segmentation has increased, its frame-level causal relationships have become more confusing, as shown in Figure1(c) and (d).

In order to render TAS be more explicable through clear causal relationships, we propose a *Causal Abstraction Segmentation Refiner (CASR)*. CASR simplifies the causality between frames, enhancing the causal explicable of segmentation results and refining various supervised baseline models for segmentation. Specifically, inspired by causal abstraction (Hu & Tian, 2022; Beckers & Halpern, 2019), we define equivalent frame-level and segment-level causal models. We retain cross-segment relationships while disregarding inter-segment relationships to simplify frames. Therefore, we whiten the feature vectors for each frame to remove inter-feature correlations and accordingly construct a frame-level causal adjacency matrix. The frames within the same segment and the frames from different segments are treated as positive and negative samples respectively, we can determine whether each frame belongs to the pre-segmentation through contrastive learning. The practical outcomes are shown in Figure1(e) and (f). After CASR, the similarity between segments decreases, and it is obviously easier to segment.

To intuitively demonstrate the causal explicable of the segmentation results, we propose a novel evaluation metric to calculates the difference between the causal adjacency matrix of the final segmentation results and the ground truth. This metric measures the causal explicable of the segmentation results. We conduct experiments by applying CASR to different backbone models, validating the generalization performance of CASR. In summary, our contributions can be summarized as follows:

- We propose a method to more clearly model causal relationships in videos, marginalizing frame-level noise relationships, thereby satisfying a consistent mapping of causal abstraction in terms of action semantics from frames to segments.

- We propose the Causal Abstraction Segmentation Refiner (CASR), enhancing causal relationships between action segments and correcting the action segmentation results of backbone models. CASR can be plugged into various backbone models.

- Our approach can boost the performance of various SOTA action segmentation models, as well as the new evaluation metrics *Causal Edit Distance* (CED) we proposed here. For example, on the 50Salads dataset, our model increases the segment edit distance of MS-TCN++ by 2.2% and that of C2F-TCN by 0.9%. On the Breakfast dataset, our model enhances the segment edit distance of ASRF by 4.6% and CETNet by 1.4%.

## 2 PRELIMINARIES AND RELATED WORKS

### 2.1 SCM & CAUSAL ABSTRACT.

**Definition 1** (Structural Causal Model (SCM) (Pearl, 2009)). *A Structural Causal Model (SCM) is a 4-tuple $\langle \mathbf{V}, \mathbf{U}, \mathcal{F}, \mathcal{P} \rangle$, where $\mathbf{V} = \{V_i \mid i \in \mathbb{N}_n\}$ are endogenous variables and $\mathbf{U} = \{U_i \mid i \in \mathbb{N}_m\}$*

are exogenous variables. Structural equations $F = \{f_i | N_i\}$ are functions that determine $\vee$ with $v_i = f_i(\boldsymbol{pa}_i, \boldsymbol{u}_i)$, where $\{\boldsymbol{Pa}\}_i \subseteq \mathbf{V}$ and $\mathbf{U}_i \subseteq \mathbf{U}$. $P(\boldsymbol{u})$ is a distribution over $\boldsymbol{u}$.

An intervention $\mathbf{V}^* \leftarrow v * (\mathbf{V}^* \subseteq V)$ acting on SCM, will obtain a new SCM with different $\mathcal{F}$. Causal abstraction can map a low-level $(SCM_L, I_L)$ to a high-level $(SCM_H, I_H)$.

**Definition 2** ($\tau$-abstraction (Beckers & Halpern, 2019)). *Let $I_L$ to be a set of interventions $\mathbf{U}_L \leftarrow u_L$ on the low-level $SCM_L = \langle \mathbf{V}_L, \mathbf{U}_L, \mathcal{F}_L, \mathcal{P}_L \rangle$, Let $I_H$ be interventions on high-level $SCM_H = \langle \mathbf{V}_H, \mathbf{U}_H, \mathcal{F}_H, \mathcal{P}_H \rangle$. Let $\tau$ be a partial function $\tau : \mathcal{D}(\mathbf{V}_L) \rightarrow \mathcal{D}(\mathbf{V}_H)$, and let $\omega : I_L \rightarrow I_H$ be $\omega_\tau(\mathbf{V}_L^* \leftarrow \boldsymbol{v}_L^*)$, where $\mathbf{V}_L^* \subseteq \mathbf{V}_L, \mathbf{V}_H^* \subseteq \mathbf{V}_H, \boldsymbol{v}_L^* \in \mathcal{D}(\mathbf{V}_L^*)$. If $\tau$ and $\omega_\tau$ are surjective and satisfy $\forall i_L \in I_L : \tau(i_L(SCM_L)) = \omega_\tau(i_L)(SCM_H)$, $(SCM_H, I_H)$ is a $\tau$-abstraction of $(SCM_L, I_L)$.*

## 2.2 RELATED WORKS.

**Causal abstraction.** Rubenstein et al. (2017)introduced the concept of *exact transformation* for the first time, using to determine when a probabilistic causal model can be transformed into another model of the same system in causal consistency. The core of *exact transformation* is the mapping $\tau$ between causal models on different levels and the surjective map $\omega$ of hard interventions. To solve the problem of ingoring non-essential differences, Beckers & Halpern (2019) further extended this concept to $\tau$-abstraction, which requires two causal abstractions with soft interventions $\tau$ to induce a specific function $\omega$. On the extremely end, there has been plenty of work make causal abstract available in much fields. On the basis of category theory, Rischel & Weichwald (2021) defines abstract $\langle \alpha = R, a, \alpha_{X^*} \rangle$ with a node set $R$ of micromodel, mapping $a$ between micromodel and macromodel, and surjective mapping set $\alpha_{X^*}$; Otsuka & Saigo (2022) defines abstract $\alpha$ by searching graph homomorphism from $\mathcal{G}_{\mathcal{M}^m}$ to $\mathcal{G}_{\mathcal{M}^M}$.

In this paper, we will use causal abstraction to demonstrate the equivalence of the frame-level and segment-level causal models our defined. In the subsequent sections, our frame-level is equivalent to the low-level and micromodel, while the segment-level corresponds to the high-level and macromodel.

**Temporal Action Segmentation.** Recently, methods based on deep learning can be mainly subdivided into those based on TCN, Transformer, and some fusion improvement methods. Many studies have introduced plugin techniques to enhance TCN-based models: GatedR (Wang et al., 2020) employed a gated forward refinement network, and Singhania et al. (2021) developed a C2F-TCN encoder-decoder model. Particularly, MS-TCN (Farha & Gall, 2019) and MS-TCN++ (Li et al., 2020) proposed multi-stage TCN to refine predictions iteratively across multiple temporal scales. Meanwhile, many improvement methods were proposed. ASRF (Ishikawa et al., 2021) adds a branch to predict segmentation point information based on MS-TCN, and SSTDA (Chen et al., 2020) integrates a self-supervised model with MS-TCN.

Comparatively, Transformer-based models have emerged as a viable alternative to TCN. AS-Former (Yi et al., 2021) uses encoder processes the video sequence and generates predictions, while the decoder takes predictions from previous layers as input. UVAST (Behrmann et al., 2022) employs a similar encoder, but predicts action segments autoregressively, effectively reducing over-segmentation. In a recent development, CETNet (Wang et al., 2023) employs a cross-enhancement transformer to efficiently learn temporal structure representations with interactive self-attention mechanisms and global and local information.

As previously noted, these studies have not focused on capturing the causal relationships between video contents, including the state-of-the-art models. Moreover, our work proposes Causal Abstraction Segmentation Refiner (CASR) on different backbone models, aiming to improve understanding and segmentation performance by simplifying causal relationships between frames.

## 3 CAUSAL RELATIONSHIP BETWEEN FRAME& SEGMENT?

In this section, we will difine the frame-level and segment-level causal models firstly, and use causal abstraction to prove they are equivalent. In light of this, we can deduce how to simplify the frame-level causal model to more accurately characterize the segment-level model, and then apply it to CASR to improve segmentation performance.

**Definition 3** (Frame-level causal model). *A frame-level causal model of a video is composed of triples $M_X = (\mathcal{S}_X, \mathcal{I}_X, \mathbb{P}_E)$, where $X = (X_i : i \in \square_x)$ is the variable set contained all of the frames, structural equation $\mathcal{S}_X$ is the set of $X_i = f_i(X, E_i)$, $\mathcal{I}_X$ is a partially ordered set of perfectly interventions, and $\mathbb{P}_E$ is the distribution of the exogenous variable $E$.*

**Definition 4** (Segment-level causal model). *A segment-level causal model of a video is composed of triples $M_Y = (\mathcal{S}_Y, \mathcal{I}_Y, \mathbb{P}_E)$, where $Y = (Y_j : j \in \square_y)$ is the variable set contained all of the frames, structural equation $\mathcal{S}_Y$ is the set of $Y_j = f_j(Y, E_j)$, $\mathcal{I}_Y$ is a partially ordered set of perfectly interventions, and $\mathbb{P}_E$ is the distribution of the exogenous variable $E$.*

In order to ensure the identifiability of frame-level and segment-level causal models, we propose hypothesis1 according to the characteristics of video data. In this way, we can satisfy the DAG structure without proposing other assumptions such as acyclic constraints.

**Hypothesis 1** (Causality Identifiability). *In the frame-level causal model, the variables $X$ in $X_i = f_i(X, E_i)$ only contains $(X^*, \leq_{X_i})$, represents there is only the former variables(frames) have causal effect to the latter variables (frames) in causal graph; likewise, the segment $Y$ in $Y_j = f_j(Y, E_j)$ only contains $(Y^*, \leq_{Y_j})$, represents there is only the former segment have causal effect to the latter segment in causal graph.*

We ensure the identifiability of the two models by assuming the causality within, then we can define the two models can be transformed. The proof process is given in the appendix A.

**Definition 5** (Exchangeability). *When frame-level and segment-level causal model satisfy the Causality Identifiability 1, the two models can be transformed to each other.*

As mentioned before, we have verified the frame-level causal model is susceptible to interference from noise terms. As shown in Figure 1(c)(d), there is a lot of redundancy between frames, but the relationships in segment-level is clearer. Because a complete action segment is represented by many different frames, so that the complete and clearer action semantic relationships in the video can only emerge at the segment-level. Therefore, when we only focus on the semantics of actions, the noise relationship in the frame-level can be marginalized. This also explains why the frame-level and segment-level causal models can be transformed into each other. In light of this, we can infer the Frame-level causal relationships:

**Corollary 1** (Frame-level causal relationships). *For any three variables $X_n^{Y_1}, X_l^{Y_1}, X_k^{Y_2}$ in the frame-level causal model $M_X$ of a video, where $X_n^{Y_1}$ and $X_l^{Y_1}$ are respectively the n-th and l-th frames in the $Y_1$th action segment, $l < n \leq T_{Y_1}$; $X_k^{Y_2}$ is the k-th frame in the $Y_2$th action segment, $k \leq T_{Y_2}$. $T_{Y_1}$ and $T_{Y_2}$ represent the whole numbers of frames in $Y_1$ and $Y_2$, and $Y_1$ occurs before $Y_2$, then*

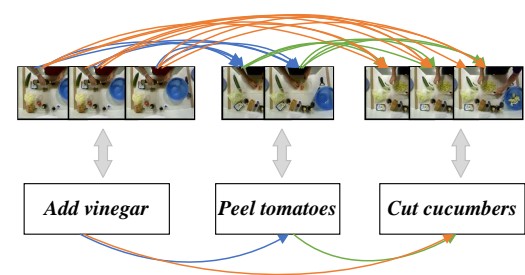

Figure 2: Causality in frame-level causal model and segment-level causal model. Assume that there are three action segments and their corresponding frames. The lines in blue represent the causal effect of "Add vinegar" to "Peel tomatoes", the lines in green represent the causal effect of "Peel tomatoes" to "Cut cucumbers", and the lines in orange represent the causal effect of "Add vinegar" to "Cut cucumbers".

- *There is no causal relationship between two frames belonging to the same action sub-segment in $M_X$, $p(X_n^{Y_1}|X_l^{Y_1}) \rightarrow 0$;*

- *Two frames do not belong to the same action sub-segment have a causal relationship in $M_X$, $p(X_k^{Y_2}|X_l^{Y_1}) \rightarrow 1, p(X_n^{Y_1} \mid X_l^{Y_1}) \rightarrow 1$.*

In this way, we only retain the relationships across action segments at the frame-level, and marginalize the inter-frame relationships within the action segments to remove irrelevant noise, as shown in Figure 2.

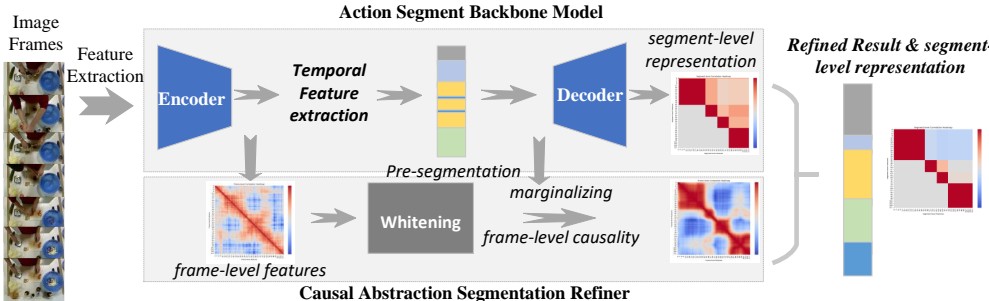

Figure 3: Overview of Causal Abstraction Segmentation Refiner (CASR). We aim to improve the action segmentation results pre-trained by the backbone model Action Segment Engineer. We extract the dimensionally reduced frame-level features, and strengthen the causal relationship between the frame-level whitening features according to the pre-segmentation results of the backbone model.

## 4 METHODOLOGY

### 4.1 OVERALL STRUCTURE

Figure 3 shows the overview of our CASR. We have defined above the frame-level and segment-level causal models can be transformed into each other, CASR aims to refine the segment-level segmentation results in the backbone model by marginalizing the causal relationships in the frame-level. For continuous frame-level features as input, we extract the features after dimensionality reduction as the input of CASR. Due to we pay attention to causal relationship between features rather than the correlation, we first batch-partition the features and then whiten them. According to the pre-segmentation representation of backbone model, we marginalize the causal relationships in frame-level, and construct frame-level causal adjacency matrix from the whitened features. In addition to the original loss function of the backbone model, we also added an additional loss function to strengthen causal representation. Under the simultaneous learning of dual branches, we comprehensively improve the segment-level segmentation results.

### 4.2 CAUSAL ABSTRACTION SEGMENTATION REFINER(CASR)

As mentioned previously, we first whiten the dimensionally reduced data features $X \in \mathbb{R}^{T \times n}$, ( $T$ is the sequence length, $n$ is the feature dimension) , and remove the correlation between features to ensure the final results we learned are causal relationships. Ermolov et al. (2021) proposed a self-supervised learning method based on latent space feature whitening, which can scatter samples in batches onto a spherical distribution through whitening to avoid feature collapse into a single point. For frame-level features with variable sequence length, in order to improve calculation efficiency, we first divide the sequence into fixed-length batches $\{x_1, x_2, \cdots, x_K\}$, $x_i \in \mathbb{R}^{L \times n}$, $L$ represents batch size, $K$ represents the number of batches divided according to the length of the video. By constraining $cov(x_i, x_i) = I$ among the features $\{x_1, x_2, \cdots, x_K\}$, perform singular value decomposition on the feature matrix to obtain the whitened feature vector $\{z_1, z_2, \cdots, z_K\}$. In order to ensure the stability of whitening, we follow the method in Ermolov et al. (2021), randomly divide the sub-batch again, and then calculate the whitening matrix independently.

Due to the first segment results usually focus on short-term features between frames, so it will affect the extraction of long-term features and even the segmentation result of the whole model. We extracted the first segmentation result from the backbone model as the pre-segmentation result, divided frame variables into positive and negative cases according to the frame-level causality derived in the previous chapter, and constructed a causal adjacency matrix. Therefore, according to the pre-segmentation result $Y = \{y_1, y_2, \cdots y_K\}, y_i \in \mathbb{R}^{L \times M}$, where $M$ is the number of action types. We extract the action type $\tilde{y}_i \in \mathbb{R}^L$ of i-th batch through the softmax function according to the pre-segmentation, and then use the broadcast mechanism to extend $\tilde{y}_i$ to $\tilde{y}_i^a \in \mathbb{R}^{L \times L}$. Thence, as mentioned in Corollary 1, we determine whether two frames belong to the same action segment frame by frame, assign the value $p(x_{T_M}^M | x_1^M)$ of the positive in the causal adjacency matrix to 0,

indicating no causal relationship in the two frames; assign the value $p(x_{T_M}^{M_1}|x_1^{M_2})$ of negative in the causal adjacency matrix to 1, indicating that there has causal effect between the two frames, thereby constructing the ground truth of the adjacency matrix

$$c_i = \tilde{y}_i^a \oplus \tilde{y}_i^b \tag{1}$$

where $c_i \in \mathbb{R}^{1 \times L \times L \times 1}$. Similarly, we use the broadcast mechanism to expand the whitened feature vector $x_i \in \mathbb{R}^{L \times n}$ from different dimensions to $\tilde{x}_i^a \in \mathbb{R}^{L \times L \times n}$ and $\tilde{x}_i^b \in \mathbb{R}^{L \times L \times n}$ respectively, then combine the whitening features of all frames in pairs to get

$$\tilde{c}_i = \tilde{x}_i^a \oplus \tilde{x}_i^b \tag{2}$$

where $\tilde{c} \in \mathbb{R}^{1 \times L \times L \times n}$, and then we utilize linear layers mapping $\hat{c}_i$ to [ 0,1] , representing the conditional probabilities between learned whitened features.

$$\hat{c}_i = sigmoid(\tilde{c}_i) \tag{3}$$

## 4.3 Causality Representation Loss

Independent of the loss function of the original backbone model Action Segment Engineer, we propose a new loss function $\mathcal{L}_{CA}$ for CASR to calculate the difference between $\hat{c}_i$ and $c_i$. The backbone model still uses its original loss function, such as the combined weighting of $\mathcal{L}_T MSE, \mathcal{L}_C E, \mathcal{L}_T R, \mathcal{L}_A L$. For the causal adjacency matrix $\hat{c}_i, c_i \in \mathbb{R}^{1 \times L \times L \times 1}$, we use contrastive learning to make the difference between frame-level causal positive pairs as small as possible and the difference between negative pairs as large as possible.

$$\mathcal{L}_{CA} = \frac{1}{KL} \sum_{i=1}^{K} (c_i - \hat{c}_i)^2 \tag{4}$$

We shrink $\mathcal{L}_{CA}$ to frame-level to balance the relationship between $\mathcal{L}_{CA}$ and the backbone model loss function. In general, the frame-level feature vector is trained by the backbone model and CASR at the same time, the loss is calculated separately, so that we can obtain the segmentation results refined by us. We will use experiments to confirm the effectiveness of our refiner and its generalization ability on various backbone models in the next section.

## 5 Experiment

In this section, we conduct sufficient experiments to answer the research questions:

- **RQ1:** How effective is CASR in improving the backbone model on downstream tasks?
- **RQ2:** How about the generalization performance of CASR applied to Action Segment Engineer of different backbone models?
- **RQ3:** Can CASR better learn the causal representation of data?

## 5.1 Settings

**Datasets.** In experiments, we use three challenging datasets: Georgia Tech Egocentric Activities (GTEA) (Fathi et al., 2011), 50salads (Stein & McKenna, 2013) and Breakfast (Kuehne et al., 2014). The GTEA dataset consists of 28 first-person perspective videos containing 7 different daily activities performed by 4 actors, and the dataset is divided into 4 splits by actors. The 50salads dataset contains the entire process of 25 people making salads, with a total of 50 videos, and is divided into 5 splits. Breakfast dataset consists of 10 cooking activities performed by 52 different actors in multiple kitchen locations. This dataset is the largest of the three datasets and is divided into 4 groups. For consistency, all videos from these datasets are set to 15 fps. We use I3D (Carreira & Zisserman, 2017) featureswhich are extracted from all frames and provided by Farha & Gall (2019).

Table 1: Refinement results based on GTEA, 50salads,and Breakfast datasets.[1]

| Methods | GTEA | | | | | | 50 Salads | | | | | | Breakfast | | | | | |
|---|---|---|---|---|---|---|---|---|---|---|---|---|---|---|---|---|---|---|
| | $F1@10,25,50$ | | | Edit | Acc | CED | $F1@10,25,50$ | | | Edit | Acc | CED | $F1@10,25,50$ | | | Edit | Acc | CED |
| MSTCN++† | 82.3 | 83.6 | 71.9 | 79.8 | 77.6 | 8.400 | 79.4 | 77.3 | 69.3 | 71.6 | 82.8 | 3.334 | - | - | - | - | - | - |
| MSTCN++† + CASR | 86.4 | 84.2 | 72.7 | 80.8 | 78.9 | 7.942 | 81.6 | 79.7 | 71.4 | 73.8 | 84.0 | 2.869 | - | - | - | - | - | - |
| Gain | 4.1 | 0.6 | 0.6 | 1.0 | 1.3 | -0.458 | 2.2 | 2.4 | 2.1 | 2.2 | 1.2 | -0.465 | - | - | - | - | - | - |
| ASRF† | 85.5 | 83.8 | 73.6 | 76.9 | 74.7 | 9.045 | 80.3 | 77.4 | 67.4 | 74.2 | 77.6 | 4.932 | 69.1 | 63.4 | 50.8 | 66.6 | 63.0 | 55.832 |
| ASRF† + CASR | 86.5 | 84.3 | 72.4 | 80.3 | 73.9 | 8.157 | 80.4 | 78.3 | 70.6 | 74.7 | 76.8 | 4.795 | 72.4 | 67.1 | 55.1 | 71.2 | 65.5 | 50.095 |
| Gain | 1.0 | 0.5 | -0.8 | 3.4 | -0.8 | -0.888 | 0.1 | 0.9 | 3.2 | 0.5 | -0.8 | -0.137 | 3.3 | 3.7 | 4.3 | 4.6 | 2.5 | -5.737 |
| CETNet† | 90.5 | 89.6 | 78.9 | 85.7 | 79.4 | 7.134 | 87.6 | 87.3 | 80.9 | 82.8 | 87.3 | 2.587 | 72.5 | 68.7 | 57.0 | 72.8 | 74.2 | 38.194 |
| CETnet† + CASR | 91.4 | 90.2 | 80.5 | 87.2 | 79.7 | 6.915 | 88.9 | 87.6 | 81.4 | 83.1 | 88.9 | 2.541 | 78.7 | 74.9 | 63.4 | 78.3 | 75.6 | 35.436 |
| Gain | 0.8 | 0.5 | 1.6 | 1.6 | 0.3 | -0.219 | 1.3 | 0.3 | 0.5 | 0.3 | 1.6 | -0.046 | 6.2 | 6.2 | 6.4 | 5.5 | 1.4 | -2.8 |
| C2F-TCN† | 88 | 86.6 | 78.3 | 81.6 | 80.6 | 7.358 | 83.5 | 81.5 | 71.8 | 75.7 | 86.9 | 2.802 | 71.6 | 68.0 | 57.1 | 68.1 | 74.6 | 49.831 |
| C2F-TCN† + CASR | 88.7 | 87.7 | 78.8 | 83.5 | 80.7 | 7.023 | 83.9 | 81.6 | 72.9 | 76.6 | 86.7 | 2.603 | 71.9 | 68.2 | 57.2 | 67.6 | 75.7 | 48.327 |
| Gain | 0.7 | 1.1 | 0.5 | 1.9 | 0.1 | -0.335 | 0.4 | 0.1 | 1.1 | 0.9 | -0.2 | -0.199 | 0.3 | 0.2 | 0.1 | -0.5 | 1.1 | -1.504 |

**Evaluation Metrics.** When evaluating action segmentation results, we use rolled-out frame-level segment labels from our CASR. For evaluation, frame-level accuracy (Acc), segmental edit distance (Edit), and segmental F1 scores with different overlapping threshold $k\%$ ($F1@k$) ($k = \{10, 25, 50\}$) are used. Acc is the most common value that reflects frame-level segmentation accuracy. Edit distance calculates the minimum number of operations required to perform a replacement operation between two frames, and measures the difference between two frames. Different overlap thresholds $k\%$ $F1$ can be used to evaluate the prediction quality of different time domain characteristics.

In order to evaluate the causal explanation ability of segmentation results, we additionally propose Causal Edit Distance (CED) to measure the difference between the adjacency matrix $\hat{C} \in \mathbb{R}^{T \times T}$ and ground truth $C \in \mathbb{R}^{T \times T}$. Smaller CED values indicate a smaller discrepancy between the causal relationships among frame-level segmentation results and ground truth.

$$CED := num(\hat{C}_{i,j} \neq C_{i,j}); i,j = 1, 2, \cdots, T \qquad (5)$$

**Baseline.** We have selected several mainstream models and state-of-the-art models as baselines, and we have introduced it before, including TCN-based method MS-TCN++ (Li et al., 2020) and C2F-TCN (Singhania et al., 2021), Transformer-based method CETNet (Wang et al., 2023), and fusion-improved methods ASRF (Ishikawa et al., 2021).

**Implementation details.** To mitigate random biases, our refiner applied to different baselines while preserving their original settings such as random seed, epochs, learnling rate. All experiments are conducted on a single GEFORCE RTX 3090. To enhance training efficiency and prevent the occurrence of degenerate matrices during whitening, we configure the batch size for frames as 512. Furthermore, following the approach outlined in Ermolov et al. (2021), we set the sub-batch size to 128.

## 5.2 QUANTITATIVE RESULTS

In order to verify the effectiveness our proposed CASR, we applied CASR to various baseline models based on different backbones, such as MS-TCN++, ASRF, CETNet, and C2F-TCN. Table 1 shows the experiment results of our method, as well as the comparison with the baseline. Since our CASR needs to be trained by adding to different methods, in order to better test our improved performance, the baseline results we display are all our reproduction results under the same experimental conditions.

Table 2: Ablation experiment result.

| Model | $F1@10,25,50$ | | | Edit | Acc | CED |
|---|---|---|---|---|---|---|
| w/ o normalization $L_{CA}$ and whitening | 77.9 | 75.5 | 67.2 | 69.9 | 81.9 | 3.298 |
| w/ o whitening | 78.9 | 76.2 | 68.1 | 71.4 | 82.2 | 3.255 |
| w/ o normalization $L_{CA}$ | 79.8 | 77.9 | 68.4 | 72.4 | 82.8 | 3.278 |
| **MSTCN++ + CASR (512,128)** | **81.6** | **79.7** | **71.4** | **73.8** | **84.0** | **2.869** |

As shown in Table 1, the segmentation performance of CASR is significantly improved when applied to different backbone models (**RQ2**), especially in terms of the causal interpretability of the model

---

[1] † represents the results is by our reproduction. The results of MS-TCN++ on the Breakfast dataset are not given because we reproduced it based on the authors' open source code, and the results obtained are far from the results published by the author (Li et al., 2020).

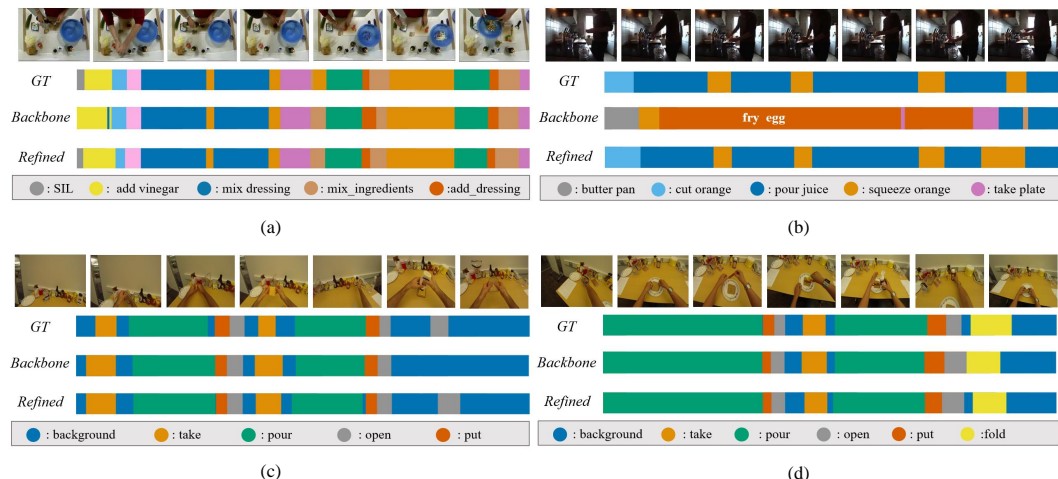

Figure 4: Segmentation results improved by CASR for different backbone models and datasets. Best view in color. (a) Correction results of MSTCN++ on the 50salads dataset. (b) ASRF correction results on the Breakfast dataset. (c) Correction results of CETnet on the GTEA dataset. (d) Correction results of C2F on the GTEA dataset.

(**RQ3**). Affected by the size of different datasets, CED measures the difference in the causal adjacency matrices of all frame-level models, so the range of its values is different under different data amounts. In cases where the amount of data is large (such as in the breakfast dataset), our segmentation performance improves significantly; in cases where the backbone model segmentation performance is relatively low (such as in MS-TCN++ and ASRF), our model performance gains also higher. (**RQ1**)

In the experiment, we whitened the frame-level feature vectors in order to remove the correlation between features and avoid all features from converging on a single point when learning conditional probabilities. At the same time, we also normalized the loss learned by CASR to prevent the loss of $\mathcal{L}_{CA}$ from affecting the recognition of the model too much. We show different results without whitening and without normalizing the loss function in Table 2 respectively, which proves that whitening does have an important impact on learning the conditional probability between frame-levels, and normalizing $\mathcal{L}_{CA}$ helps to balance the relationship with the original loss function of the baseline model to improve segmentation performance.

Table 3: Different batch size in experiment result.

| (Batch size, Sub-batch size) | $F1@10, 25, 50$ | | | Edit | Acc | CED |
|---|---|---|---|---|---|---|
| (512,64) | 75.4 | 72.5 | 61.9 | 70.5 | 79.3 | 3.719 |
| (256,128) | 76.6 | 74.2 | 65.2 | 69.0 | 81.1 | 3.502 |
| (256,64) | 75.3 | 73.4 | 64.2 | 68.4 | 78.8 | 3.982 |
| (128,64) | 77.1 | 74.6 | 65.2 | 69.9 | 80.8 | 3.453 |
| **Ours (512,128)** | **81.6** | **79.7** | **71.4** | **73.8** | **84.0** | **2.869** |

As previously mentioned, we reset the batch size and sub-batch size from the frame-level to improve the efficiency of whitening and constructing the causal adjacency matrix. So we tested different batch sizes and sub batch sizes respectively, and the results obtained are shown in Table 3. That is why we chose the combination (512,128) in our experiment.

### 5.3 QUALITATIVE RESULTS

As shown in Figure 4(a), obviously, CASR can correct over-segmentation errors. We have refined the phenomenon of the backbone model incorrectly identifying a small segment of other actions in one action segment, as well as incorrectly identifying the dividing points of adjacent action segments. Figure 4 (c) and (d) also show that CASR can identify some action segments not recognized by the backbone model, especially action segments with short duration, which refines the identification omission problem of the backbone model.

Figure 4(b) shows the process of making orange juice from a third-person perspective. As shown in the figure, the backbone model obviously misidentified the main action segment as an action

unrelated to the video content, such as "fry egg". This may also be due to the low brightness of the video. CASR can correct such out-of-context misrecognition, demonstrating CASR has excellent ability to identify and characterize the semantics of action segments.

## 5.4 DISCUSSION.

During the experiment, we found that CASR may have a problem of solidification of causal relationships. In the case where several action segments have nothing to do with each other, the segmentation results obtained may be unsatisfactory. As shown in Figure 5(a)(b), in the ground truth, the action followed by "peel cucumber" is "add dressing" with low correlation, but because "peel cucumber" and "cut cucumber" have extremely high correlation , so after the pre-segmentation result learns the "peel cucumber" action segment, some frames are divided into "cut cucumber". Since CASR has a strong dependence on pre-segmentation results, our CASR amplifies this causal relationship based on backbone, which instead leads to incorrect segmentation. Therefore, we hope to improve the solidified causal relationship in the next step of work. In this paper, we ignore the causal relationship between all frames within an action segment in order to enable the segment-level to represent action semantics more clearly, causing us to also ignore some fine-grained differences that may distinguish similar actions. Therefore, in the next work, we will consider the frame-level causality within the segment and design some new indicators to calculate the connection between frames. When this value exceeds a certain threshold, the two frames in one action segment is considered to have a causal relationship from front to back.

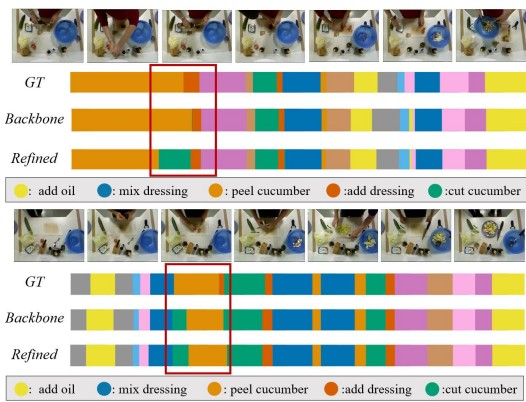

Figure 5: Causal relations in frame-level causal model and segment-level model. Let us take a short video in 50salads as an example. Assume that there are three action segments and their corresponding frames. The lines in blue represent the causal effect of "Add vinegar" to "Peel tomstoes",the lines in green represent the causal effect of "Peel tomstoes" to "Cut cucumbers", and the lines in orange represent the causal effect of "Add vinegar" to "Cut cucumbers".

## 6 CONCLUSION

In this paper, we enhance the explainability of temporal action segmentation tasks from a causality perspective. Our focus is on how to remove frame-level noise and simplify the frame-level causal model. To this end, we propose a method to marginalize the noise relationship of frame-level causal models, introduce CASR to improve the performance of different backbone segmentation models, and propose a new evaluation metric CED to verify its causal interpretability. The core of CASR is to convert the causal relationship of the frame-level model to a segment-level with a clearer causal relationship based on the pre-segmentation results of the action segment engineer, and propose a new loss function to learn the segment-level causal model so that each frame can be determined whether it belongs to its pre-segmentation. We have proven the effectiveness and generalization ability of CASR in a large number of experiments. In the future, we will build more interpretable models under various assumptions, improve the current possible problem of solidification of causal relationships, and reduce reliance on pre-segmentation results.

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

## A  PROOF OF EXCHANGEABILITY BETWEEN CAUSAL MODELS

We have proposed the frame-level causal model and segment-level causal model of video can be transformed into each other in Definition 5. We will prove the definition here.

*Proof.* Let $\mathcal{M}_X = (\mathcal{S}_X, \mathcal{I}_X, \mathbb{P}_{E,F})$ be a liner frame-level causal model over the variables $W = (W_i : 1 \leq i \leq n)$ and $Z = (Z_i : 1 \leq i \leq m)$ with

$$\mathcal{S}_X = \{W_i = E_i : 1 \leq i \leq n\} \cup \{Z_i = \sum_{j=1}^{n} A_{ij} W_j + F_i : 1 \leq i \leq m\} \tag{1}$$

$$\mathcal{I}_X = \{\varnothing, do(Z = z), do(W = w, Z = z) : \omega \in \mathbb{R}^n, z \in \mathbb{R}^m\} \tag{2}$$

and $(E, F)$ $\mathbb{P}$ where $\mathbb{P}$ is any distribution over $\mathbb{R}^{n+m}$ and $A$ is a matrix. Assume that there exists an $a \in \mathbb{R}$ such that each column of $A$ sums to $a$. Consider the following transformation that averages the $W$ and $Z$ variables:

$$\tau : \mathcal{X} \to \mathcal{Y} = \mathbb{R}^2 \tag{3}$$

$$\begin{pmatrix} W \\ Z \end{pmatrix} \longmapsto \begin{pmatrix} \widehat{W} \\ \hat{Z} \end{pmatrix} = \begin{pmatrix} \frac{1}{n} \sum_{i=1}^{n} W_i \\ \frac{1}{m} \sum_{j=1}^{m} Z_j \end{pmatrix} \tag{4}$$

Futher, let $\mathcal{M}_Y = (\mathcal{S}_Y, \mathcal{I}_Y, \mathbb{P}_{\hat{E},\hat{F}})$ over the variables $\{\widehat{W}, \hat{Z}\}$ be a segment-level causal model of video with

$$\mathcal{S}_Y = \left\{ \widehat{W} = \hat{E}, \hat{Z} = \frac{a}{m} \widehat{W} + \hat{F} \right\} \tag{5}$$

$$\mathcal{I}_X = \{\varnothing, do(\hat{W} = \hat{\omega}), do(hatZ = \hat{z}), do(hatW = \hat{\omega}, hatZ = \hat{z}) : \hat{\omega} \in \mathbb{R}, \hat{z} \in \mathbb{R}\} \tag{6}$$

$$\hat{E} \frac{1}{n} \sum_{i=1}^{n} E_i, \hat{F} \frac{1}{m} \sum_{i=1}^{m} F_i \tag{7}$$

Then segment-level $\mathcal{M}_Y$ is an exact $\tau$-abstraction of frame-level $\mathcal{M}_X$.

