# OpenReview forum: "CASR: Refining Action Segmentation via marginalizing frame-level causal relationships"
_ICLR.cc/2024/Conference — ICLR 2024 Conference Withdrawn Submission_

### Official Review · Reviewer_F2hH · 2023-10-23

**Soundness:** 2 fair
**Presentation:** 1 poor
**Contribution:** 2 fair
**Rating:** 3
**Confidence:** 3

**Summary:**

This paper proposes a action segmentation method based on causal relationships, aiming to enhance segmentation performance and improve the model's interpretability by capturing causal relationships. The author first introduces the concept of causal abstraction to formalize frame-level causal relationships. Then, CASR, a video action segmentation enhancer based on causal abstraction that can be used with different baseline models, is introduced. The effectiveness and generalization performance of the method are validated on three different datasets.

**Strengths:**

1. The proposed method improves the segmentation performance of different baseline models on three benchmark datasets for action segmentation.
2. The proposed method has good reproducibility and the code is provided.
3. The strategy of causal abstraction is interesting.

**Weaknesses:**

1. The writing of the paper is poor. The methods section is not presented clearly enough. There are more cases of undefined and inconsistent variable names in the paper. And there are more obvious grammatical problems in some places. In addition, some parts of the paper are still in latex form.
2. The paper's experiments don't support the authors' ideas very well.
3. Clearly, the textual description in Figure 5 is inconsistent with the information conveyed by the image. The textual description in Figure 5 appears to be a minor extension of the description in Figure 2.

**Questions:**

1. For the content in Weakness 3, I would like the authors to provide an explanation and consider revising the description in Figure 5.
2. For the content in Weakness 2, I listed the following questions.
- First, I think the authors need to describe the experimental setup of Tables 2 and 3 more clearly. The experiments covered in Tables 2 and 3 should include the following key information: first, make it clear on which dataset the experiments were conducted; second, make it clear which model was chosen in Table 3. These details are important for the reader to understand the context of the experiment and the plausibility of the results.
- Second, concerning the combination of batch size and sub-batch size (512,128) in Table 3, while this can improve the efficiency of whitening and constructing the causal adjacency matrix, it remains unclear why enhancing the efficiency of these two would have an impact on the segmentation performance of the model. I expect the authors to provide a more detailed explanation to help readers better understand this correlation.
- Third, I think the performance improvement of the four selected baseline models is not enough to reflect the generalization and robustness of the proposed method. The authors should provide more robust experimental data and may consider separate experimental validation of the TCN-based (e.g., MS-TCN++, ETSN, ASRF), Transformer-based (e.g., AsFormer, CETNet, LTContext) and Diffusion-based (e.g., DiffAct) methods respectively.

---

### Official Review · Reviewer_jAKX · 2023-10-25

**Soundness:** 4 excellent
**Presentation:** 1 poor
**Contribution:** 3 good
**Rating:** 6
**Confidence:** 3

**Summary:**

In order to make Temporal Action Segmentation (TAS) more explicable and enhance the segmentation ability, this paper explores frame-level causal relationships by proposing a method of marginalizing frame-level noise relationships and a Causal Abstraction Segmentation Refiner (CASR). It retains cross-segment relationships while discarding inter-segment ones, which satisfies a consistent mapping of causal abstraction in terms of action semantics from frames to segments. Besides, contrastive learning is used to identify whether each frame belongs to the corresponding segment. In addition, a new evaluation metric, Causal Edit Distance (CED) is proposed to evaluate the causal explanation ability of segmentation results. The experimental results show that the proposed method can boost the performance of various SOTA action segmentation models.

**Strengths:**

1. The proposed method of marginalizing frame-level noise relationships can effectively handle the noise emerged by frame-level causal relationships and make it feasible to suggest macro action relationships through frame relationships.
2. The paper leverages contrastive learning to enhance the segmentation performance and the proposed method can be plugged into various backbone models.
3. The proposed metric, CED can verify the causal interpretability of segmentation results well. And the experimental results demonstrate that the approach in this paper can boost the performance of various SOTA action segmentation models.

**Weaknesses:**

1. The notations and formulations in this paper are a little confusing that some notations are not clarified and some with the same implication are expressed in different forms.
2. Some expressions are not precise and the whole paper needs further proofreading.

**Questions:**

1. The notations in this paper need carefully checking. For example, at the end of the 3rd line of Definition 1, "functions that determine V", the form of notation V is different from the preceding expression. In addition, in the 4th line of Definition 1, pa and Pa seem to express the same meaning, so the notations need to be unified. Besides this, there are also some other notation issues that need to be checked.
2. At the end of the 2nd line of Definition 4,  "Y is the variable set contained all of the frames", it seems that Y is the variable set contained all of the segments not the frames.
3. Why is there a question mark after the title of Section 3 CAUSAL RELATIONSHIP BETWEEN FRAME& SEGMENT? Besides, some section titles have punctuations after them, while others do not. Please ensure a consistent format.
4. Please proofread the paper carefully again to avoid the issues in formatting and expression.

---

### Official Review · Reviewer_RzwV · 2023-10-29

**Soundness:** 2 fair
**Presentation:** 1 poor
**Contribution:** 2 fair
**Rating:** 3
**Confidence:** 3

**Summary:**

This paper proposes to utilize causal abstraction for temporal action segmentation by modelling frame-level and segment-level relationships. The proposed method is evaluated on three common benchmark datasets.

**Strengths:**

Modelling frame-level and segment-level relationships is an interesting topic for temporal action segmentation.

**Weaknesses:**

1. The motivation for using causal reasoning for temporal action segmentation is unclear to me. Why should we study an interpretability topic for this specific task instead of more general tasks? The contribution is more at the application level instead of more insightful advancements.

2. The writing and presentation need to be improved.
- The flow of the introduction section and the abstract can be improved. This also makes the first weakness more evident.
- Some concepts and terms are used without definition, such as SEM, the noise term, and Action Segment Engineer.
- Figure 1 needs to be improved. The texts are too small. The connection between this figure and the causal relationship is unclear to me. Why are (b)(d)(f) asymmetric?

3. The experiments should include ASFormer, which is an important backbone for action segmentation.

4. In Table 1, apart from the reproduced results, the original results of the previous method should also be reported. The reproduced results are much lower than the original results. What is the reason for this? This might undermine the effectiveness of the proposed method.

**Questions:**

Please refer to weaknesses, especially weakness 4.

---

### Official Review · Reviewer_xRrB · 2023-11-01

**Soundness:** 2 fair
**Presentation:** 1 poor
**Contribution:** 1 poor
**Rating:** 1
**Confidence:** 3

**Summary:**

The paper presents a method to improve temporal action segmentation by introducing an causal abstraction segmentation refiner (CASR). CASR enhances the causal abstraction model by suppressing noisy relationships between frames. The proposed method is validated on three datasets: GTEA, 50salads and Breakfast. The experimental results show that the proposed method can be applied to different models for improving temporal action segmentation performance.

**Strengths:**

The proposed method can be applied to different models to enhance causal abstraction modelling as shown in Table 1. The ablation study validates that the whitening operation and the contrastive loss helps improve performance.

**Weaknesses:**

The proposed method is not well formulated in the framework of the causal model.
Definitions/collary/hypothesis are provided in Sections 2 and 3, but some terms and notations are not clearly presented.
It is not clear about how the proposed method is incorporated into the causal model and why it works.

The experiments are not sufficient to validate the effectiveness of the proposed method. No comparisons of the propsoed method and state-of-the-art mehtods are provided.

The paper is poorly written. The idea is not clearly presented. It is difficult to understand Figures 1 and 2 even after reading the corresponding text in the paper.
Also, there are some gramatical errors which further increases the difficulty of understanding the paper.

**Questions:**

See the Weaknesses section.